# Efficient Design of Thin Wall Seating Made of a Single Piece of Heavy-Duty Corrugated Cardboard

**DOI:** 10.3390/ma14216645

**Published:** 2021-11-04

**Authors:** Berta Suarez, Luisa M. Muneta, Gregorio Romero, Juan D. Sanz-Bobi

**Affiliations:** Mechanical Engineering Department, Escuela Técnica Superior de Ingenieros Industriales, Universidad Politécnica de Madrid, C/José Gutiérrez Abascal, 2, 28006 Madrid, Spain; luisa.mtzmuneta@upm.es (L.M.M.); gregorio.romero@upm.es (G.R.); juandedios.sanz@upm.es (J.D.S.-B.)

**Keywords:** composite sandwich structures, thin-walled structures, anisotropic material, corrugated core, homogenization approach, first-order shear deformation theory, FSDT, FEM simulation, finite element analysis, design process

## Abstract

Corrugated cardboard has waved cores with small flutes that prevent the use of detailed numerical models of whole structures. Many homogenization methods in the literature overcome this drawback by defining equivalent homogeneous plates with the same mechanical behaviour at a macro-mechanical scale. However, few homogenization works have considered complete structures, focusing mainly on beams or plates. For the first time, this study explores the application of homogenization approaches to larger structures as an aid in their design process. We also considered triple-wall boards rather than single- and double-wall configurations commonly addressed in the literature. To this end, we adapted the homogenization methods proposed by Talbi and Duong to analyze thin-walled stools made of triple-wall corrugated cardboard. Using a progressive design process, we performed an efficient stool design by removing material zones with lower stresses, with 35% less material, 35% lower vertical deflections, and 66% lower stresses than the initial design. Unlike other corrugated cardboard stools, this design comprises just one folded piece instead of three, thus saving storage space. These results demonstrate the utility of homogenization techniques as an aid in the design process of whole structures made of corrugated cardboard. Further research will consider buckling analysis.

## 1. Introduction

Finite element analysis (FEA) greatly facilitates the design process of many products, avoiding the construction of failed prototypes. Concerning products made of corrugated cardboard, this advantage is not so evident since it is inexpensive and easy to handle, so that prototypes have low economic and time costs. In this paper, the authors aim to show that FEA can also be very useful when designing products made with this material. The main advantage is not to avoid prototyping, but to guide the design stages towards more efficient solutions. Likewise, it could help to choose the most suitable type of cardboard for each product, avoiding the need to gather an extensive assortment of materials to test different prototypes.

In this work, we applied FEA to a piece of furniture made of corrugated cardboard to achieve a more efficient design. To define the material properties, we adapted the homogenization methods proposed by Talbi [1] and Duong [2], as described in Section 2.3.

Conventional furniture designs often rely on traditional knowledge in handicraft manufacturing. Moreover, their structural elements are often intentionally oversized. However, FEA becomes an essential tool when dealing with unconventional furniture made of thin-wall structural elements. In [3,4,5,6,7,8,9], we can find some studies on the FEA of wood furniture. Other previous research studies also considered other materials, such as laminated bamboo [10], honeycomb cardboard [11], corrugated cardboard [12] or fibre-reinforced concrete [13].

### 1.1. Corrugated Cardboard

Corrugated cardboard is a material for everyday use, light, economical and sustainable. In addition to packaging, it can have other uses, such as construction and indoor furniture [14,15,16]. Its high strength-to-weight ratio makes it ideal for furniture manufacturing, though a careful design is needed to ensure rigidity.

It presents a sandwich structure with small waves in the intermediate layers (Figure 1), called fluting. Flutings are glued to flat sheets of paper, called liners, with a water-resistant starch-based adhesive [17]. Liners support bending loads, and flutings support transverse shear, helping to stabilize the former by resisting out-of-plane deformations [18,19]. In this way, the mechanical properties of liners and flutings are efficiently combined [20], providing a higher stiffness-to-weight ratio than an equivalent solid panel made of any of the individual constituent materials [21]. Liners are usually made of softwood kraft pulp to provide strength, with grammages ranging from 125 to 440 g/m^2^, while flutings have lower grammages, from 80 to 180 g/m^2^ [15,17,20]. Boards can present various wall configurations: single-sided, with only one fluting and one liner, and single-, double- and triple-wall (Figure 1), with the strength increasing with the number of plies.

Flutings are classified by their height and the number of flutes per unit length. Table 1 shows the most common flute types, designated as A, B, C, E or F, the C flute being the most commonly used for boxes. There are other less common flute types, such as D, with a height of 2 mm; G, thinner than 0.55 mm; K, thicker than 5.0 mm; and even a thinner flute, called O [15,22]. These letters were assigned according to their introduction into the market, having no relation to their size [17]. Larger flutes provide greater vertical strength and cushioning, while smaller flutes enhance graphic capabilities and structural integrity.

In paperboard manufacture, cellulose fibres tend to align in the flow direction, called machine direction (MD) [15]. The perpendicular direction on the paperboard surface is called cross direction (CD). Corrugated cardboard has the same manufacture direction that paperboard [24], MD being perpendicular to the principal axes of the corrugations and CD parallel to them (Figure 2). Then, both paper and corrugated cardboard are orthotropic materials, with better mechanical properties in MD than in CD [15,25].

Being a low-cost, lightweight, and environmentally friendly material, the use of corrugated cardboard for packaging has steadily increased in the past decade [19,26]. The global production of packaging paper and board increased from 193 to 256 million tons between 2008 and 2018 [27]. This effect was also influenced by the growth of online commerce [28]. In 2020, the global demand for containerboard was 69 million tons, 40% of the global demand for paper [29]. In 2018, the recycling rate for paper and cardboard packaging in the EU was 83% [30] of waste material. Waste cardboard can be used in its original form, but it can also be used in new composite materials [28].

Due to its great strength-to-weight ratio, excellent burst strength and resistance to crushing, corrugated cardboard is also suitable for furniture manufacture. However, a careful design is needed to ensure rigidity [31]. Thus, a good understanding of its mechanical behaviour is required to use it in an optimum way. Many previous studies have focused on the properties of corrugated cardboard and how the external environment affects its performance [20,32,33,34,35,36]. The mechanical properties of various types of liners and flutings in MD and CD can be found in [1,17,24,25,33,37,38,39,40,41,42,43,44,45,46,47,48,49].

### 1.2. Thin-Wall Furniture

Based on its thickness, we can classify the structural elements of furniture as ultrathin, below 10 mm; thin, from 10 to 15 mm; standard, from 16 to 19 mm; thick, from 20 to 40 mm; and ultra-thick, above 40 mm [50]. Thin-wall furniture, made of thin or ultrathin structural materials, is a current trend in furniture design [51]. It is usually made of wood composite panels, such as plywood, particleboard, or medium-density fibreboard (MDF), which can be laminated with other materials [50]. Due to its light weight, it can be considered a good alternative for trade shows and conventions. It can even be a suitable option for students or professionals with upward mobility, who will probably move often.

When dealing with the design of thin-wall furniture, a structural calculation is of particular relevance [50]. In addition to the strength requirements imposed on the materials, a second challenge lies in the joints between different panels [28,51]. Thin-walled structures can also exhibit buckling and warping problems, extensively studied in the scientific literature. Some analytical, numerical, and experimental studies on the buckling analysis of thin-wall beams can be found in [52,53]. Other studies on the buckling of corrugated cardboard structures can be found in [37,44,46,54,55].

In this work, our objective was to design a thin-wall furniture piece made of a different material, such as heavy-duty corrugated cardboard, whose sandwich structure could provide the required strength. Compared to wood composites, it has the advantage of being foldable. Thus, it requires fewer joints. Being low-cost and easy to transport and mount, in addition to the applications mentioned above, it can also be considered to meet the needs for accommodation in improvised shelters for emergencies [56].

#### Corrugated Cardboard Furniture

Corrugated cardboard furniture is usually made of pieces that could be flat-packed and assembled at home, using folds, slots and tabs. In the early 1960s, Peter Murdoch designed the Spotty chair [57], a flat-pack disposable chair that could be assembled simply by folding it in shape. In the early 1970s, Craig Hodgetts, Robert Mangurian, and Keith Godard designed Punch-Out [31], a low-cost furniture line made of heavy-duty corrugated cardboard, with flat pieces that even children could assemble to form their own tables and chairs. Today, many specialized companies [58,59,60,61,62,63,64,65,66,67,68,69] offer a great variety of corrugated cardboard furniture (such as chairs, armchairs, tables, shelves, beds, standing desks or podiums) [70], to be used at home, the office or trade shows. Many freelance designers also present their designs of corrugated cardboard furniture in design and architectural social media platforms or blogs [71,72,73].

As evidence of the growing interest in this type of furniture, the Japanese bedding company Airweave [74] provided 18,000 and 8,000 high-resistance cardboard beds for Olympians and Paralympians at the 2020 Tokyo Olympics [75,76]. They were conceived as a recycling initiative and were intended to be converted into other paper products. They will be reused for COVID-19 patients in a temporary medical facility in Osaka [77].

Another use of waste corrugated cardboard, as part of lightweight multi-layered panels with alternating plies of corrugated cardboard and veneer, was examined in [28]. Their study, considering different types of end corner joints between rigid panels, confirmed the suitability of this material for furniture and interior applications.

### 1.3. Homogenization Techniques

Different approaches can be used to analyze the strength of corrugated cardboard products: experimental [78]; analytical [79,80]; analytical-numerical [81,82,83] or purely numerical [33,84,85,86]. Due to the small size of the fluting, numerical methods are inadequate to analyze any structure made with this material on a micromechanical scale. Instead, we may use homogenization approaches. They allow considering its sandwich structure as a homogeneous plate [87,88], providing almost as accurate responses for homogenized models as for real structures [89].

Some homogenization techniques use analytic methods to obtain the engineering constants of the equivalent material [48,90,91,92,93]. Others apply the classical laminate theory (CLT) or the first-order shear deformation theory (FSDT) [1,2,45,94,95] to obtain the stiffness matrix of an equivalent plate [1,2,19,45,96,97,98,99]. Others use FEA of a representative volume element (RVE) to find an equivalent homogeneous plate [43,100,101,102,103,104].

Most homogenization studies centre on isolated flutings or single-wall corrugated boards, though some of them also consider double-walled corrugated panels [41,94,98,105,106,107]. Moreover, most of the existing literature on corrugated cardboard models focuses on homogenization methods, with few practical applications in actual designs.

### 1.4. Scope of the Study

This work aims to apply FEA for the structural calculation of corrugated cardboard furniture as an aid in its design process. As an example, we chose a stool made of this material to show the effectiveness of this method. This paper shows the process we followed to design the stool, performing a structural calculation of each intermediate design to assess its validity. In a future study, we also intend to consider a buckling analysis of the different design stages. However, this is beyond the scope of this work.

## 2. Materials and Methods

### 2.1. Design Stages

As a starting point, we based the first design on the geometry of a commercial stool, the so-called Kenno Stool [108,109] (Figure 3), designed by the Finnish designer Heikki Ruoho [110,111]. We chose this model for its simplicity. It comprises three pieces assembled perpendicularly, forming a closed structure that can be used as either a stool or a low table. It has a trapezoidal shape, resting on the ground, indistinctly, either on the wide or narrow part of the trapezoid. It has two vertical sidewalls with a vertical groove in the middle of their upper side. They are placed parallel to each other and covered by a third piece, whose ends fit into the groove of the former pieces.

We slightly increased its dimensions, since the original stool was conceived for children. We also replaced the original honeycomb cardboard with heavy-duty triple-wall corrugated cardboard [112,113,114,115], with which we obtained excellent results in a previous study of cardboard seating [12]. A 1970s child’s chair design from the hplusf design lab was made with this material. It was called Punch-out [116] and was temporarily exhibited at the MoMA [117]. Today, some contemporary furniture manufacturers, such as Chairigami (USA) [118] or Konno Konpou (Japan) [119], also use this material.

We applied FEA to this design, using a homogenization approach to characterize the mechanical properties of corrugated cardboard. In Section 2.3 and Appendix A, we present a thorough description of the homogenization technique used in this work.

From the numerical analysis performed, we obtained the deflections and stresses of this stool under some applied loads, according to the European Standards EN 1728 [120] and EN 12520 [121], both applicable to seating designs.

We then modified this design by removing both side panels. Therefore, the second design consisted of a single piece that the final user could fold for storage (Figure 4).

We lengthened the ends of the cardboard panel towards the opposite face and crossed them to ensure the structural strength of the stool. To maintain the total width of the top/bottom face, we placed the crossing point near it. We also reduced the width of one end to insert it into a slot made at the opposite end. The stool should also have two grooves on the top/bottom surface for inserting both ends, preventing them from moving. We also analyzed this design under the same load conditions.

To achieve more significant savings in material and storage space, we even opened the stool downward by removing the lower face (Figure 5). We now crossed both ends at an intermediate height inside the stool. However, it could rest only on the edges that limit the open surface, having a single possible position, unlike the previous designs. 

Next, we modified the design by cutting both ends of the stool directly from the front and rear walls, opening a hole in those walls and folding the cut material inward (Figure 6a). This design saves even more material and storage space, since its ends could be placed inside the cut walls again. The angle formed between the front/rear wall and the seating surface should be the same as the angle between the ends and the seating surface, since both pieces should have the same length. During preliminary simulations, high longitudinal displacements were found at the bottom edges. Hence, we closed it on the bottom side by extending the front/rear walls to the bottom (Figure 6b) and connecting them.

Taking into account the orthotropic behaviour of corrugated cardboard, we analyzed each design for two material orientations: orientation I, with MD (*x*-axis) parallel to the folding lines, supposed to provide higher bending stiffness, and orientation II, with CD (*y*-axis) parallel to the folding lines, supposed to ease the folding process. We also considered two different body orientations: the wide part of the trapezium facing up and down.

The results thus obtained clearly show the utility of FEA, even for products made of an inexpensive and easy-to-handle material such as corrugated cardboard.

### 2.2. Finite Element Models

To develop the FE models of the stool designs, we used commercial software that includes a specific module for the structural analysis of composite materials.

We modelled the stool as a layered linear elastic shell. To do so, we combined the shell elements with a layered linear elastic material suitable for orthotropic laminates. In this way, the program applied the FSDT formulation internally. As input data, we introduced the stiffness matrices of the inner liners, the outer liners, and the fluting, together with the thickness and material model of each layer of the sandwich panel. We also used solid elements to model the loading pad used to apply loads on the seating surface.

We defined the contact conditions between intersecting panels using a mapped mesh defined so that two intersecting panels share the shell nodes lying on their intersection line. To define the contact between the solid elements of the loading pad and the shell elements of the stool panels, we used a multiphysics coupling provided by the commercial software; specifically, we used a solid-thin structure connection for this purpose.

Finally, we performed a static analysis with each model.

In the following sections, we define the FE model in more detail.

#### 2.2.1. Geometry

All the designs considered had a seating surface 380 mm long and 400 mm wide and a height of 400 mm (Figure 7).

We built finite element (FE) models of all stool designs using homogenized shells with a mapped mesh (Figure 8) made up of square elements approximately 5 mm long. The number of boundary elements used in the models shown in Figure 8 ranges from 1704 for model (f) to 3810 for model (b).

The angle α between the top and front/rear panels was modified from 70° to 90°, with a 5° step, preserving the length of the seating surface. We considered the fourth design with α = 90° just for comparison, since it could rotate around the edges formed by the top and front/rear panels, thus being unstable. Figure 9 gathers the geometry variations for the fourth design to show where the board ends intersect the seating surface.

#### 2.2.2. Material

The material considered for all designs was a heavy-duty triple-wall A-flute corrugated cardboard. Its homogenized properties were defined in the FE model using a layered material with seven layers: 1 and 7 are outer liners, 3 and 5 inner liners, and 2, 4, and 6 flutings (Figure 10). For each layer, we introduced either the liner thickness or the fluting height, together with its homogenized stiffness matrix, previously computed as described in Appendix A.

We used the engineering constants of the constituent materials reported in [45] to compute the stiffness matrices, since they have high elastic moduli and would provide high bending stiffness. Table 2 shows the engineering constants, E_i_, G_ij_, and ν_ij_. They are given in the lamina reference frame, with the 2-axis parallel to the CD, and the 1-axis parallel to the MD.

To model the height and period of the fluting, we took the values indicated in the Tri-Wall Pak patent [114] for A flutes. We also took the thicknesses stated in [114] for the liners. For the fluting, we considered the grammage of 150 g/m^2^ specified in [113], corresponding to a thickness of 0.25 mm. Table 2 also shows the thickness of the liners and fluting, t, and the height, h, and period, P, of the fluting, all taken from the references mentioned above.

We also considered two orientations: I, with MD (red *x*-axis) parallel to the folding lines, and II, with CD (green *y*-axis) parallel to the folding lines (Figure 11).

#### 2.2.3. Loads and Constraints

We applied the load distribution defined in the Eurocode EN 1728 [120], using a cylindrical loading pad, placed 175 mm from the front edge of the seat and centred on the width of the seating surface (Figure 12).

We modelled the pad as a solid steel cylinder with 180 mm diameter, covered with a 10 mm layer of polyurethane foam, using a free tetrahedral mesh. We applied a vertical force of 1300 N, according to Eurocode EN 12520 [121], for domestic seats. It was uniformly distributed on the upper surface of the cylinder and transmitted to the shell through a multiphysics coupling.

We applied simply supported boundary conditions at the lower edges of the folded panels (Figure 13). We restricted the three displacements of the lower front edge, but only the lateral, y, and vertical, z, displacements of the lower back edge (shown in blue).

### 2.3. Homogenization Approach

In this study, we applied a homogenization approach based on the first-order shear deformation theory (FSDT). It is an evolution of our previous work [12], which was in turn based on previous research by Talbi [1] and Duong [2].

The stiffness matrix of any lamina of a laminate can be easily formulated in the lamina reference frame, 123. However, to use a common reference system, we need to express the stiffness matrices of all laminas in the global laminate reference frame, xyz. This process is straightforward for liners, since they are flat, but not for flutings. Due to their waved shape, the material parameters for each section differ from the laminate reference frame, xyz, to the lamina reference frame, 123, in which they are known [87] (see Figure 14). Thus, we need to change the reference system of the stiffness matrix of the flutings.

Berthelot [55] applied a similar method to composite materials by rotating around the *z*-axis, normal to the laminate. For corrugated materials, however, the rotation has to be performed around the *y*-axis, or CD. Talbi [1] and Duong [2] performed this change of reference system to formulate their homogenization methods for single- and double-wall corrugated cardboard panels, respectively. Once the stiffness matrix of the fluting was transformed, they applied the FSDT to simplify the constitutive equations. Then, they integrated the stresses through the whole laminate thickness to get the internal forces, *N* and *T*, and the bending moments, *M*. After the integration, the *z* coordinate disappeared from the formulation, reducing the problem’s dimensionality from 3D to 2D. Then, they performed a second integration along the MD over a fluting period to obtain the average values. In this way, they expressed the generalized constitutive law as follows.
(1)$$\left[\begin{array}{c} N\\ M\\ T \end{array}\right]=\left[\begin{array}{ccc} A & B & 0\\ B & D & 0\\ 0 & 0 & H \end{array}\right]\cdot \left[\begin{array}{c} \varepsilon_{m}\\ \kappa \\ \gamma_{s} \end{array}\right]\;$$

$\varepsilon_{m}$ is the membrane strain vector, κ the curvature vector, and $\gamma_{s}$ the transverse shear strain vector. *A* is the extensional stiffness matrix, *D* the bending stiffness matrix, B the bending-extension coupling stiffness matrix and *H* the transverse shear stiffness matrix. These matrices can be used to model a homogenized shell. For small structures, such as beams or plates, FE analysis can be performed analytically. However, when dealing with larger structures, an FE code is needed. Some FE packages include the FSDT formulation and directly work with the *A*, *B*, *D*, and *H* matrices. If it is not included, we can use the expressions found in the literature for the engineering constants of the homogenized shell as functions of these matrices [19,122].

In a previous work [12], we also applied this homogenization method. We computed the *A*, *B*, *D*, and *H* matrices outside the FE model and introduced them into the FE model. However, no additional information concerning the thickness and number of laminas was needed to perform the analysis. Since the FE model had no information to undo the homogenization after the simulation, the results of the analyses were averaged over the laminate thickness, and we needed to post-process them.

In this work, we used a different approach to avoid this post-processing, thus facilitating the graphical representation of the simulation results. As before, we changed the reference system to express the stiffness matrices of the corrugated layers in the laminate reference frame. Unlike before, this time, we directly introduced these matrices into the FE model. However, since they depend on the x-coordinate, they need to be processed before being introduced into the FE model. Thus, we performed a similar integration to that made by Talbi and Duong, but not on the *A*, *B*, *D*, and *H* matrices, but on the stiffness matrix of the corrugated layers. To do so, we first averaged each matrix through the *z*-coordinate and then over the x-direction, or MD (see Appendix A).

We then introduced the stiffness matrix of each layer into the FE model. We used a specialized module for composite materials that includes a layered linear elastic material model, which internally performs a second homogenization through the thickness of the whole laminate. It is based on the FSDT, like the methods of Talbi and Duong. This time, the total number of laminas and their respective thickness had to be introduced into the FE model. Then, it had the necessary information to undo the homogenization after the simulations. In this way, the results directly show different stress fields for each lamina, instead of just an average value, with no further post-processing.

The main drawback of this method is that it cannot be performed with basic FE packages but only with specific modules for composite materials. In return, we could simplify the calculation of the stiffness matrices while increasing the precision of the results of the FE analysis. Unlike before, any change in the number of sandwich layers or their thickness can be made directly inside the FE model, keeping the same stiffness matrices. Only when we want to change the geometry of the corrugated layers, we would need to recalculate their stiffness matrices outside the FE model. Using an FE module specialized in composite materials, this methodology also allows one to change the orientation of the corrugated panel and even to consider different orientations for individual layers inside the panel. If desired, it is also possible to perform delamination studies.

## 3. Results and Discussion

### 3.1. Homogenized Material Properties

Table 3 gathers the nonzero elements of the stiffness matrices computed for each layer of the corrugated board in the laminate reference frame, using Voigt notation. The fluting has lower values than the liners, since it is mainly void.

### 3.2. Parametric Study for α = 70° to 90°

For the four stool designs, we performed a parametric variation of α, from 70° to 90°, with a step of 5°. The influence of α found in the vertical deflections for the first, second, and third designs is very low. Similarly, its influence on the longitudinal deflections is also low for the first design. Figure 15 shows the longitudinal deflections for the second and third designs. For the second design, they decrease with increasing α, while for the third design, they increase with increasing α.

Figure 16 shows the vertical and longitudinal deflections found for the fourth design. The former have a minimum for α = 80° and the latter for α = 75°. However, since the vertical deflections are better for α = 80°, we consider this the best angle.

Figure 17 shows the stress distributions *σ**_xx_* and *σ**_yy_* in the global reference system for the fourth design. For both orientations, *σ_xx_* and *σ_yy_* also present a minimum for α = 80°.

### 3.3. Analysis of Designs with α = 80°

#### 3.3.1. First Design for α = 80°

This paragraph shows the results obtained for the first design with α = 80° (see Figure 7), considered the best angle from the parametric analysis. Deflections u, v and w, are respectively aligned with the global *x*-, *y*- and *z*-axes (see Figure 13).

Figure 18 shows the vertical deflections, w, for designs with bottom and top discontinuities and both ply orientations.

For the designs with bottom discontinuity, the vertical deflections show a revolution geometry about the vertical axis, with a flat bottom. Their maximum values for top discontinuity are located on the seating surface panel closest to the load application area.

For the designs with bottom discontinuity, they are 11% lower for orientation II. However, for the designs with top discontinuity, they are 23% lower for orientation I, which provides a higher bending stiffness. They are lower for the designs with top discontinuity. They show an 82% reduction for orientation I from bottom to top discontinuity and a 74% reduction for orientation II. We can explain this reduction by the span length of the seating surface, which has a single panel for bottom discontinuity, but is divided into two panels with half the span length for top discontinuity.

For both discontinuities, the longitudinal deflections are lower for orientation II. In any case, the four designs analyzed show small values, below 1 mm.

Since the material is orthotropic, we should not use von Mises stresses. Figure 19 and Figure 20 respectively show the components *σ**_xx_* and *σ**_yy_* of the stress tensor in the laminate reference frame.

For the designs with bottom discontinuity, *σ**_xx_* and *σ**_yy_* are distributed mainly on the seating surface. For orientation I, *σ**_yy_* is also transmitted to the front and rear panels. On the contrary, for orientation II, *σ**_xx_* and *σ**_yy_* are transmitted to the side panels.

For the designs with top discontinuity, the maximum stresses were found on the panels covering the sidewalls, specifically at the vertical ends inserted into the side panels’ slots. Figure 21 shows the stress distribution for *σ_xx_* and *σ_yy_* for the designs with top discontinuity again, but now removing the front panel of the seating surface, thus revealing the stress distribution in such central panels, with the maximum stresses shown in dark red and dark blue.

According to the maximum stress criterion [123], applicable to orthotropic materials, the maximum values of *σ**_xx_* and *σ**_yy_* should be lower than the tensile strength of the constituent materials in the MD, *σ_t_*_,MD_, and in the CD, *σ_t_*_,CD_, respectively (see Figure 11).

The tensile strength of structural paper can vary from 17 to more than 75 MPa in MD and from 9 to 35 MPa in CD [124]. In this study, we considered as reference values the tensile strengths found in [124] for a base paper with similar elastic moduli that the constituent materials of the analyzed stool: *σ_t_*_,MD_ = 75.4 MPa in MD and *σ_t_*_,CD_ = 22.7 MPa in CD. For these limit values, all the configurations analyzed meet the maximum stress criterion. Moreover, even for other materials with tensile strengths quite close to the lower limit of the stress ranges indicated above, the stresses obtained would be above the limit values.

#### 3.3.2. Second Design for α = 80°

This paragraph presents the results found for the second design with α = 80°. Figure 22 shows the vertical deflections for designs with bottom and top discontinuities and both ply orientations.

For orientation I, the distributions of vertical deflections show a geometry of revolution about the vertical axis, but they have an almost cylindrical shape for orientation II.

They are lower for orientation I and top discontinuity, showing a 92% reduction (from 14.44 to 1.07 mm). The improvement due to orientation is substantially more significant than for the first design, with 57% and 65% reductions for configurations with bottom and top discontinuities, respectively. Regarding the discontinuity location, for the top position, we found 82% and 78% improvements for orientations I and II, respectively.

Longitudinal deflections range from 0.3 to 1.4 mm. They are also lower for orientation I and top discontinuity. For both orientations, the highest values are in the middle-upper part of the front inner panel.

Figure 23 and Figure 24, respectively, show the stress distributions *σ**_xx_* and *σ**_yy_*.

For the bottom discontinuity, the maximum stresses concentrate in the central area of the seating panel. For orientation I, they are transmitted to the front and rear panels. However, for orientation II they are transmitted to the lateral edges of the seating surface.

Both *σ_xx_* and *σ_yy_*, are below *σ_t_*_,MD_ (75.4 MPa) and *σ_t_*_,CD_ (22.7 MPa), thus complying with the maximum stress criterion. Moreover, they would also be valid for any other structural paper, whose tensile strengths in MD and CD are, respectively, higher than 17 and 9 MPa.

#### 3.3.3. Third Design for α = 80°

The stresses and vertical deflections are similar for the third design and the second design with bottom discontinuity. However, the longitudinal displacements are somewhat higher due to the removal of the lower panel. This effect was also shown in preliminary studies for the fourth design, with high longitudinal displacements of the lower rear edge (see Figure 5). Thus, we reintroduced the lower panel in the fourth design, since it prevents relative sliding between the front and rear lower edges.

#### 3.3.4. Fourth Design for α = 80°

This paragraph presents the results found for the fourth design with α = 80°.

Figure 25 shows both the vertical and longitudinal deflections for ply orientation I.

The trend showing lower vertical deflections for orientation I than for orientation II is maintained. For orientation I, they show an almost trapezoidal shape. For orientation II, they show an almost cylindrical shape placed on the front side of the stool, with its axis oriented from side to side. Maximum vertical and longitudinal deflections for orientation I are 0.6 and 1.2 mm, respectively, these being quite low.

Figure 26 shows the stress distributions *σ**_xx_* and *σ**_yy_* in the global reference system.

Besides the seating surface, there are other higher stresses at the intersection of the inner panels with the seating surface and on the folding lines at the lower edge of the front and rear panels, which appear to act as stress concentrators. This effect can be seen as a consequence of using less material. However, the stresses in these zones are quite below the tensile stresses. So, they do not pose any problem, at least from a static point of view.

Both *σ_xx_* and *σ_yy_* are much lower than *σ_t_*_,MD_ (75.4 MPa) and *σ_t_*_,CD_ (22.7 MPa), thus fulfilling the maximum stress criterion. They would also be valid for any other structural paper, whose tensile strengths in MD and CD are, respectively, higher than 17 and 9 MPa.

#### 3.3.5. Comparative Results for α = 80°

Figure 27 and Figure 28 show the vertical and longitudinal deflections for all designs.

The vertical deflections for orientation II are higher in the second than in the first design due to the elimination of the side panels. In contrast, for orientation I, they are of the same order of magnitude. In the worst case, the vertical deflections increase 206% (from 4.7 to 14.4 mm), from the first to the second design, while in the most favourable case, they increase by 22% (from 0.9 to 1.1 mm). We can explain this behaviour as the combination of the antagonistic effect of two factors: on the one hand, the negative effect of eliminating the side panels, especially in designs with orientation II, in which the stresses of the seating surface were transmitted to that panel, and on the other hand, the positive effect of introducing a new supporting system, with inner triangular structures. Although this second effect is beneficial for both orientations, it cannot overcome the negative effect of the other factor, especially in designs with orientation II.

For the best configurations of all the designs analyzed, the vertical deflections are lower for the fourth design, being 0.6 and 1.1 mm, respectively, for orientations I and II. Compared to the first design with top discontinuity, with vertical deflections of 0.9 and 1.2 mm, they are reduced by 33% and 8% for orientations I and II, respectively. Compared to the second design with top discontinuity, with deflections of 1.1 and 3.1 mm, they are reduced by 45% and 64%. Compared to the third design, with deflections of 6.0 and 13.8 mm, they are reduced by 90% and 92%. These reductions are lower with respect to the first and second designs because it was not possible to consider any configuration with top discontinuity in the third design.

It is remarkable that after removing a significant amount of material, the vertical deflections for the fourth design are even lower than for the first design with top discontinuity. This effect is due to the high efficiency of the inner panels added when removing the side panels from the first design, since they have a triangular structure, whose effectiveness is well known. Additionally, the results obtained for the fourth design are even better than those found for the second and third designs, which already included the inner panels. This behaviour is due to a better distribution of the intersection lines of the inner panels with the seating surface in the fourth design. The inner panels divide the seating surface into two halves, acting as intermediate supports in the second and third designs. However, they divide it into three zones in the fourth design, acting as two intermediate supports, thus reducing the span and, consequently, the maximum vertical deflections.

The longitudinal deflections are higher for the fourth than for the first and second designs. However, they are kept within reasonable limits of 1.2 mm for orientation I.

Figure 29 and Figure 30 respectively show the stress distributions *σ**_xx_* and *σ**_yy_*.

For orientation II, *σ_xx_* and *σ_yy_* are lower for the fourth design than for any other design. We can see the same trend for orientation I, except for the second design with top discontinuity.

### 3.4. Summary Results

For the best configurations of each design for α = 80°, Table 4 gathers the area, A, the vertical, w, and longitudinal, u, deflections and the stresses along MD, *σ_xx_*, and CD, *σ_yy_*.

Figure 31 shows the variation of these parameters compared to the best results of the first design, expressed as a percentage of the corresponding value in the first design.

In terms of deflections, the best results correspond to the fourth design, with orientation I and α = 80°, with maximum vertical and longitudinal deflections of 0.6 and 1.2 mm, respectively. There is a noticeable improvement compared to the first design, from which it evolved, whose best results are maximum vertical and longitudinal deflections of 0.9 and 0.2 mm, respectively. These results lead to a 33% reduction for the vertical deflections, but a 500% increase for the longitudinal deflections. Despite the high increase for the longitudinal deflection, its maximum value is just slightly above 1 mm for the fourth design. Moreover, the fourth design has an area 34% lower than the first design.

We can extract the following conclusions regarding different aspects of the possible configurations of the stool design:Ply orientation. The vertical and longitudinal deflections are lower for orientation I, except for the first design, with slightly lower values for orientation I with bottom discontinuity.Discontinuity location. The vertical and longitudinal deflections are lower for designs with top discontinuity, because the seating surface is divided into two different panels with half the span of the whole seating surface.Bottom panel. We should keep the bottom panel, because it prevents longitudinal sliding between the lower edges of the front and rear panels.α angle: In the first three designs, α has little influence on the vertical deflections. However, in the fourth design, the lowest vertical deflections correspond to an intermediate angle of 80°. The best results correspond to those angles leading to a more uniform distribution of the seating surface. That is, for those designs with the inner panels dividing the seating surface into three zones of equal length, so that none of them tends to present more significant deflections than the others (see Figure 8).

## 4. Conclusions

### 4.1. Main Findings

It is known that corrugated cardboard has higher bending stiffness for orientation I. In this work, we quantified this improvement for real applications. For vertical deflections, it ranges from 23% to 65%, finding with the best results for orientation I, except for the first design with bottom discontinuity, with slightly better results for orientation II.

For orientation I, the first design sidewalls show low stresses and can be removed.

As expected, the triangular structures inside the stool improve its static behaviour.

In the first and second designs with top discontinuity, the seating surface is divided into two parts with half the span of the seating surface. This division leads to a more favourable configuration than the corresponding designs with bottom discontinuity.

We should keep the bottom panel, since it prevents any longitudinal sliding between the lower edges of the front and rear panels.

The edges where the inner triangular structure contacts the seating panel act as intermediate supports. The seating surface has two intermediate supports in the fourth design, but only one in the second. Since configurations with more supports are most favourable, the fourth design has better static behaviour than the second and third designs.

### 4.2. Concluding Remarks

Corrugated cardboard has a great strength-to-weight ratio, excellent burst strength and resistance to crushing, thus being an ideal material for furniture manufacture. However, a careful design is needed to ensure rigidity [31]. This work aimed to apply numerical methods for the structural analysis of corrugated cardboard furniture, as an aid in their design process, to obtain efficient designs with the best resistance-to-cost ratio.

As an example, we chose a stool made of heavy-duty triple-wall A-flute corrugated cardboard. We performed static analyses on various stool designs, with a geometric evolution guided by the stresses found in previous design stages. The selection of this specific type of furniture has no particular relevance, being just a way to show the feasibility and benefits of numerical analysis in the design practice of corrugated cardboard furniture.

To define the mechanical properties of corrugated cardboard, we used a homogenization approach based on the first-order shear deformation theory (FSDT). It is an evolution of our previous work [12], which was in turn based on prior research by Talbi [1] and Duong [2]. Together with [12], a novelty of this work, is applying a homogenization technique to the numerical analysis of whole structures made of corrugated board, thus extending the scope of previous studies, usually limited to beams and plates. Although the analysis of simple structures, such as beams or plates, can be performed analytically, more complex structures, such as those considered in this work, should be studied with numerical techniques, such as FEA. A second novelty is the possibility of analyzing multiple-wall panels of any number of layers, in addition to the single- and double-wall configurations commonly addressed in the literature, also broadening the scope of previous works.

We computed the stiffness matrix of an equivalent homogeneous plate for each fluting, first averaging over the laminate thickness and then along the MD. To model the whole board, we inserted these stiffness matrices of flutings and liners into a FE model, using a layered material model based on the FSDT. Unlike other previous works, this methodology provides a way to easily model multiple-wall boards, since the homogenized matrices are independent of the number of plies of the laminate. In this way, the number of plies and the thickness of the liners can easily be changed inside the FE model.

We then performed a static analysis. The starting design of the stool evolved to three other designs, taking into account the deflections and stresses found in the FEA. Together, we analyzed four different designs under the load conditions defined by the Eurocodes EN 1728 [120] and EN 12520 [121] for seating. The first design, based on the geometry of a commercial stool made of three panels assembled in perpendicular directions, forming a closed structure, was chosen because of its simplicity. We found zones with lower stresses and progressively removed some of them. We also included an inner triangular structure to compensate for removing the side panels from the initial design.

The fourth design has higher strength than the others, showing the lowest vertical deflections and stresses, with reductions of 44% for w, 90% for *σ**_xx_*, and 67% for *σ**_yy_* compared to the starting design. It also requires 44% less material, thus reducing material costs. It is also made from a single foldable piece, requiring less storage space and reducing the possibility of losing pieces when stored. Therefore, it is significantly more efficient than the first design, based on its static behaviour, the amount of material needed and the required storage space. However, we do not discuss aesthetic or ergonomic aspects.

As expected, the results of this study demonstrate the utility of homogenization techniques as an aid in the design process of whole structures made of corrugated cardboard. The proposed methodology can be applied to the design process of any other piece of furniture, such as a shelf, a bed, a desk, or any other structural element made of corrugated cardboard. It can help to optimize its design by choosing an optimal geometry for a given material. It can also help to choose the most suitable material for a predefined geometry, by comparing panels with different numbers of walls. In both cases, it would lead to material savings. FE models can also be used to analyze delamination or buckling situations and take corrective actions when needed. These potential situations should be considered in future research. Comparative fatigue analyses would also be interesting [3].

## Figures and Tables

**Figure 1 materials-14-06645-f001:**
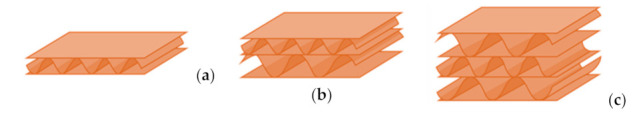
Board styles: (**a**) single-wall; (**b**) double-wall; (**c**) triple-wall. Reprinted with permission from ref. [12]. Copyright 2021 Elsevier.

**Figure 2 materials-14-06645-f002:**
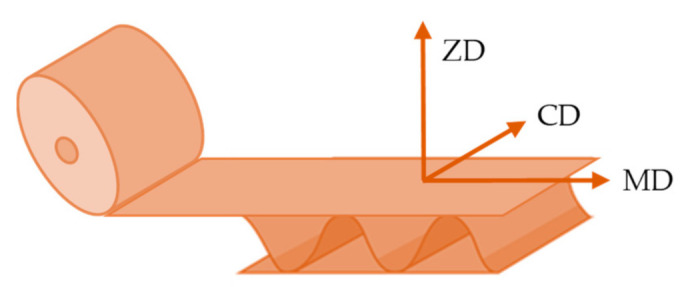
Machine direction (MD), cross direction (CD) and through-thickness direction (ZD). Reprinted with permission from ref. [12]. Copyright 2021 Elsevier.

**Figure 3 materials-14-06645-f003:**
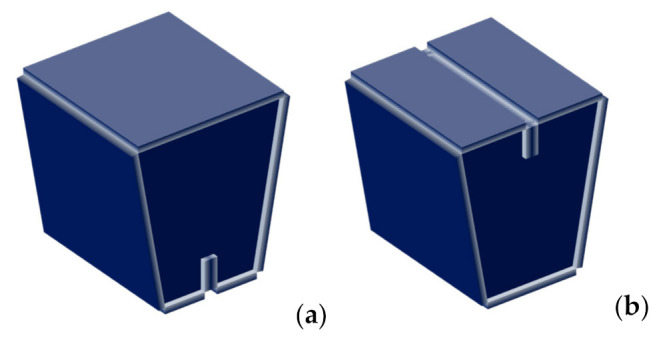
1st design with: (**a**) bottom discontinuity; (**b**) top discontinuity.

**Figure 4 materials-14-06645-f004:**
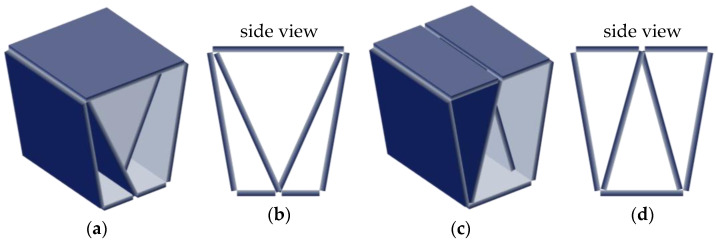
2nd design with: (**a**,**b**) bottom discontinuity; (**c**,**d**) top discontinuity.

**Figure 5 materials-14-06645-f005:**
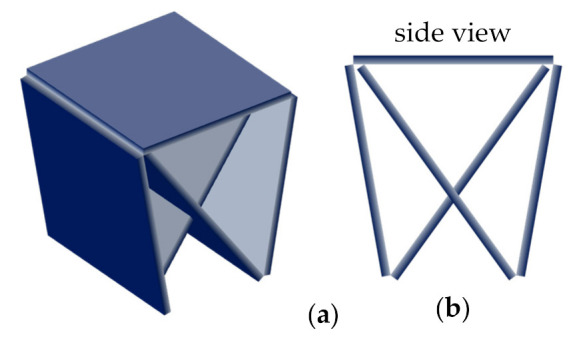
3rd design: (**a**) perspective view; (**b**) front view.

**Figure 6 materials-14-06645-f006:**
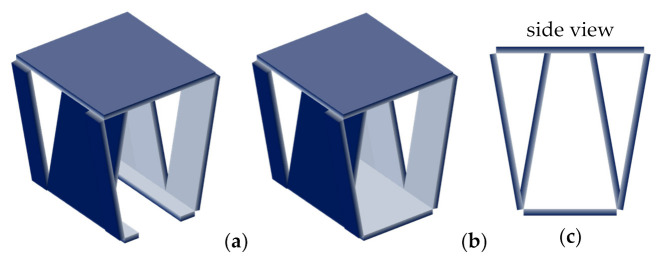
4th design: (**a**) preliminary open design; (**b**) final closed design; (**c**) side view.

**Figure 7 materials-14-06645-f007:**
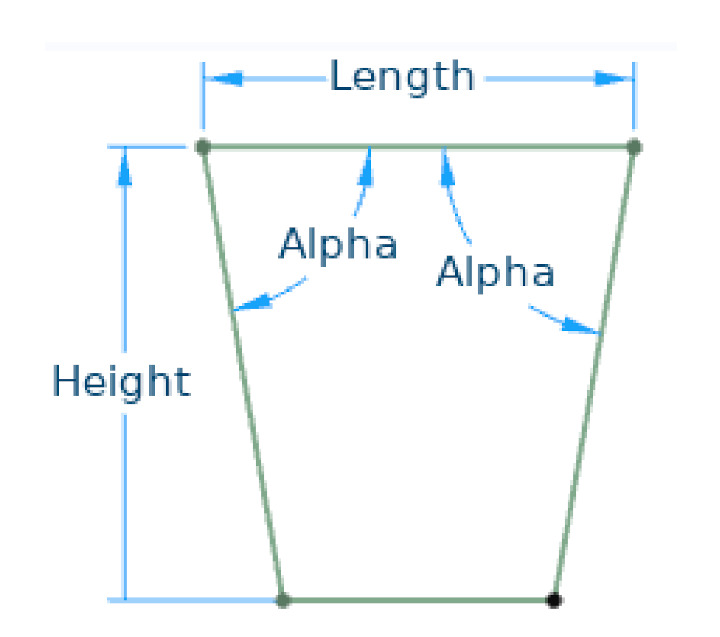
Model main dimensions.

**Figure 8 materials-14-06645-f008:**
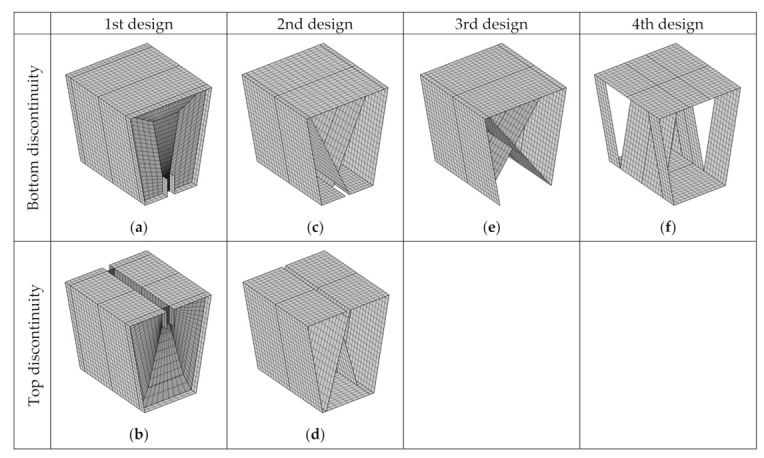
Mapped meshes: (**a**,**b**) 1st design with bottom/top discontinuity; (**c**,**d**) 2nd design with bottom/top discontinuity; (**e**) 3rd design; (**f**) 4th design.

**Figure 9 materials-14-06645-f009:**
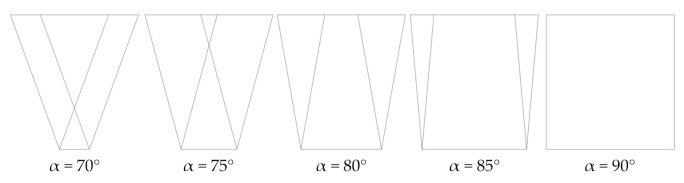
Geometry variations for the 4th design.

**Figure 10 materials-14-06645-f010:**

Layered material.

**Figure 11 materials-14-06645-f011:**
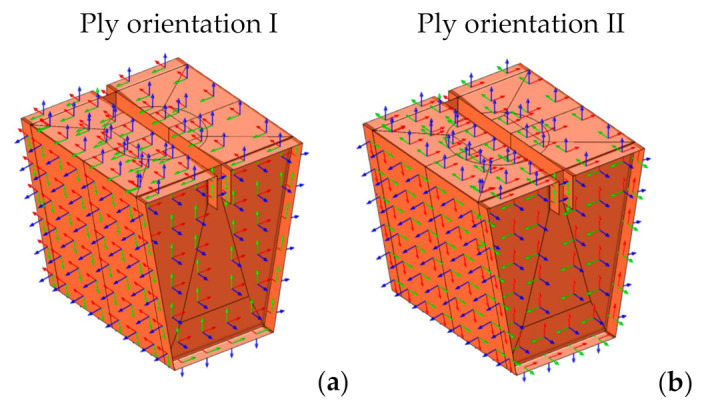
1st design with top discontinuity (red: MD; green: CD). Ply orientation: (**a**) I; (**b**) II.

**Figure 12 materials-14-06645-f012:**
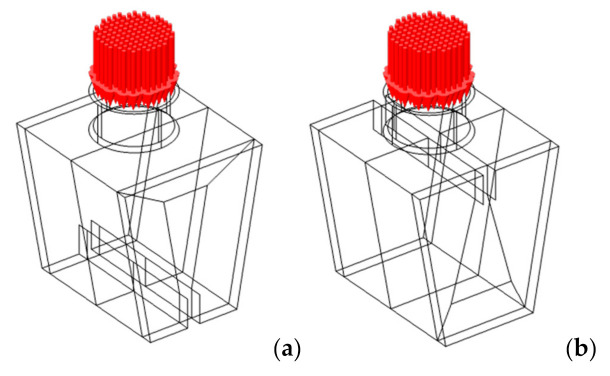
Loads applied to the 1st design with bottom (**a**) and top (**b**) discontinuity.

**Figure 13 materials-14-06645-f013:**
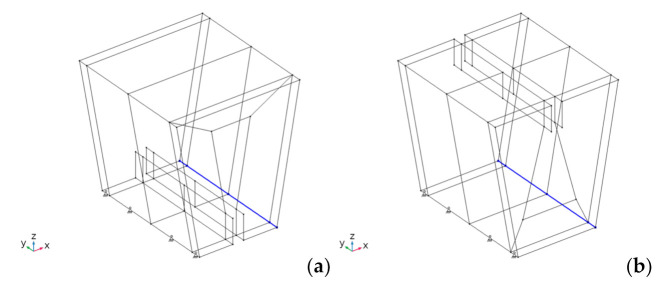
Boundary conditions applied to the 1st design with bottom (**a**) and top (**b**) discontinuity.

**Figure 14 materials-14-06645-f014:**
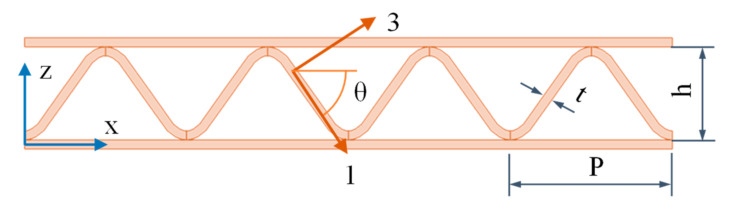
Corrugated lamina showing the laminate, xyz, and lamina, 123, reference frames. Reprinted with permission from ref. [12]. Copyright 2021 Elsevier.

**Figure 15 materials-14-06645-f015:**
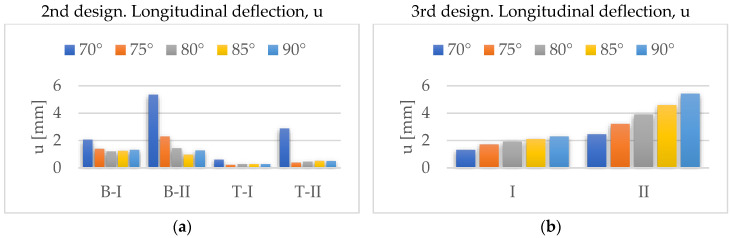
Evolution of maximum longitudinal deflections, u, with α, for designs: (**a**) 2nd; (**b**) 3rd.

**Figure 16 materials-14-06645-f016:**
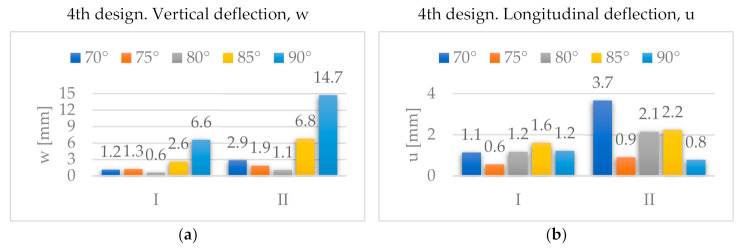
Evolution of maximum deflections with *α*, for the 4th design: (**a**) w; (**b**) u.

**Figure 17 materials-14-06645-f017:**
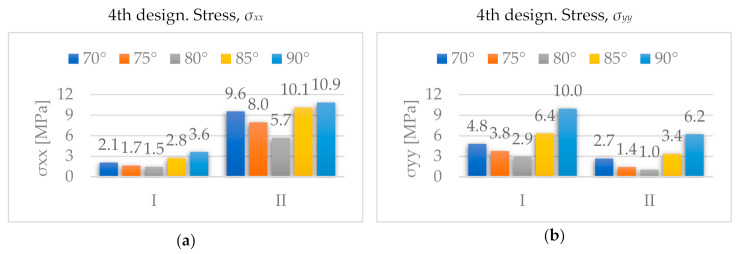
Evolution of maximum stresses [MPa] with *α*, for the 4th design: (**a**) *σ**_xx_*; (**b**) *σ**_yy_*.

**Figure 18 materials-14-06645-f018:**
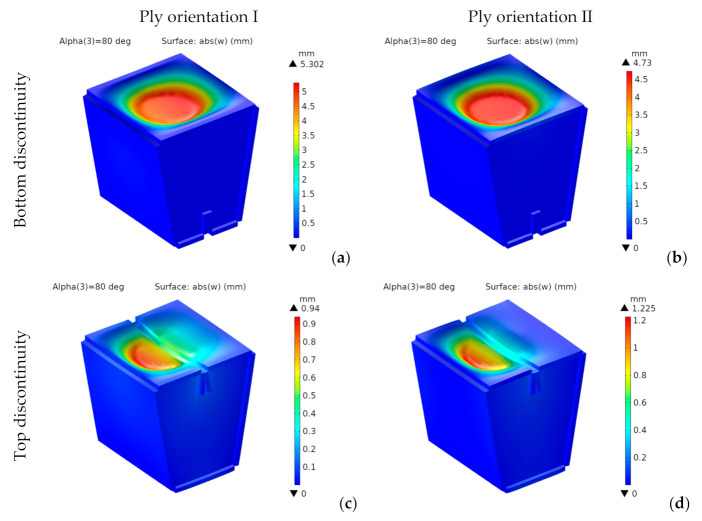
1st design. Vertical deflection [mm] for bottom (**a**,**b**) and top (**c**,**d**) discontinuity.

**Figure 19 materials-14-06645-f019:**
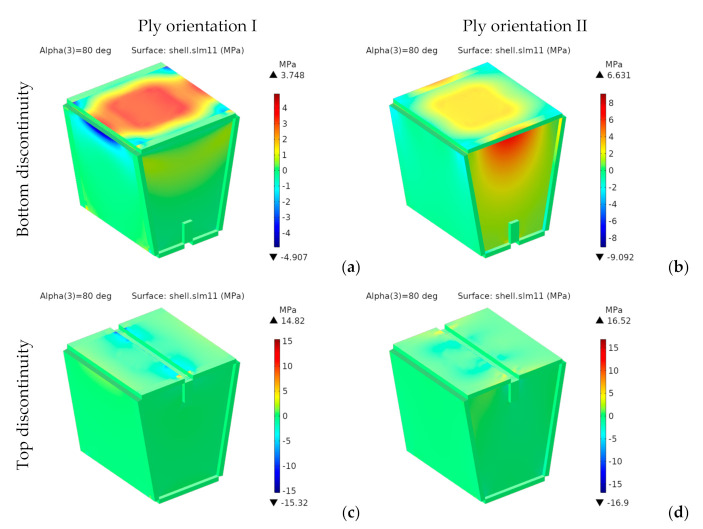
1st design. Stress *σ_xx_* [MPa] for bottom (**a**,**b**) and top (**c**,**d**) discontinuity.

**Figure 20 materials-14-06645-f020:**
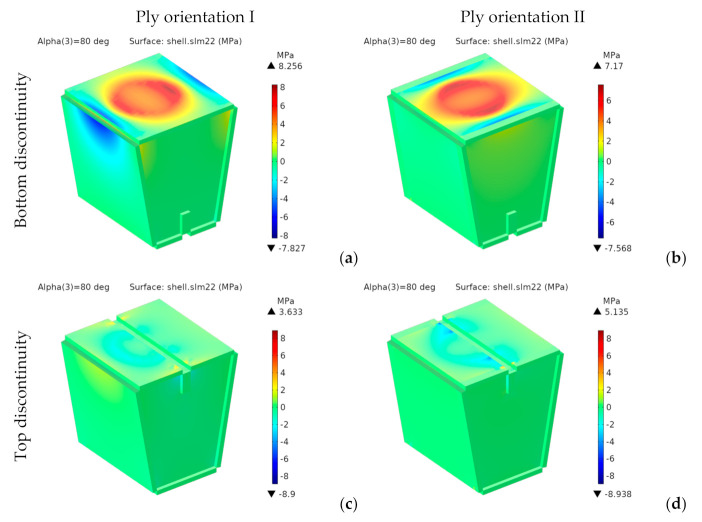
1st design. Stress *σ_yy_* [MPa] for bottom (**a**,**b**) and top (**c**,**d**) discontinuity.

**Figure 21 materials-14-06645-f021:**
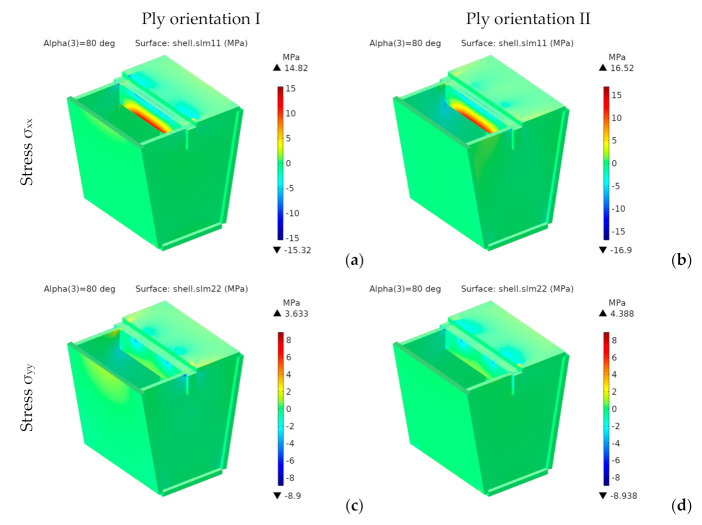
1st design. Stresses under the seating surface for top discontinuity: (**a**,**b**) σ_xx_ [MPa], (**c**,**d**) σ_yy_ [MPa].

**Figure 22 materials-14-06645-f022:**
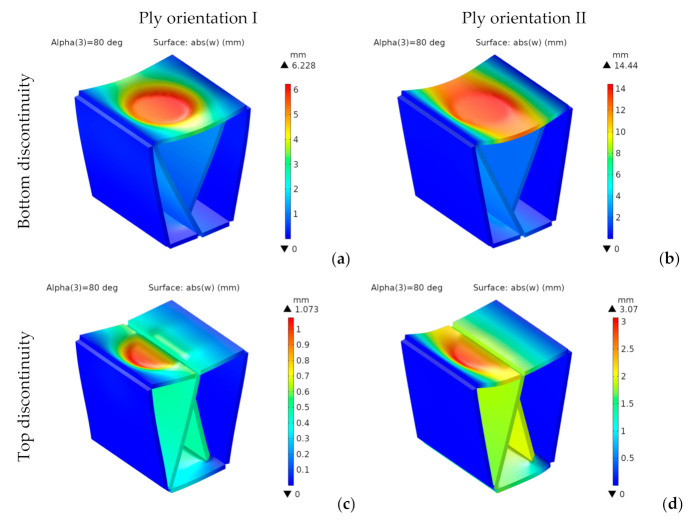
2nd design. Vertical deflection [mm] for bottom (**a**,**b**) and top (**c**,**d**) discontinuity.

**Figure 23 materials-14-06645-f023:**
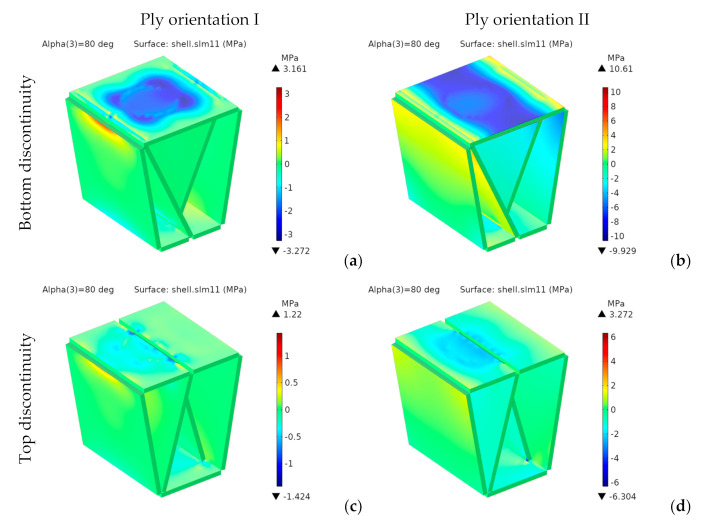
2nd design. Stress *σ_xx_* [MPa] for bottom (**a**,**b**) and top (**c**,**d**) discontinuity.

**Figure 24 materials-14-06645-f024:**
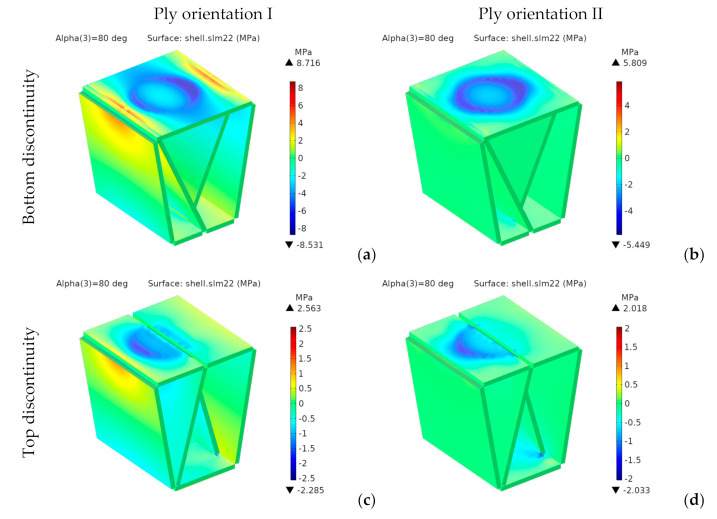
2nd design. Stress *σ_yy_* [MPa] for bottom (**a**,**b**) and top (**c**,**d**) discontinuity.

**Figure 25 materials-14-06645-f025:**
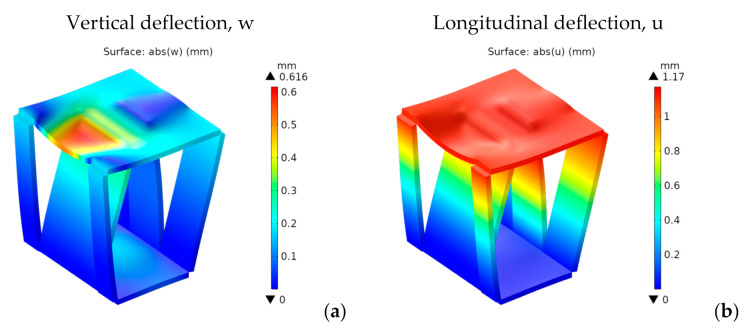
4th design. Deflections for ply orientation I [mm]: (**a**) vertical, w; (**b**) longitudinal, u.

**Figure 26 materials-14-06645-f026:**
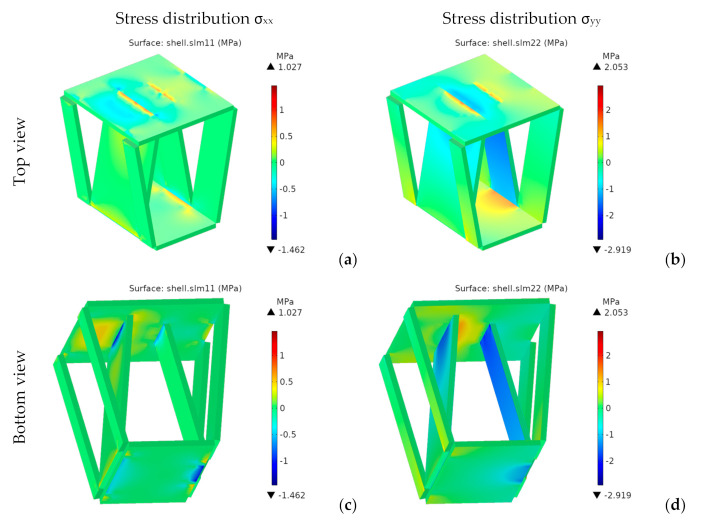
4th design. Stress for ply orientation I [MPa] (**a**,**c**) *σ_xx_*; (**b**,**d**) *σ_yy_*.

**Figure 27 materials-14-06645-f027:**
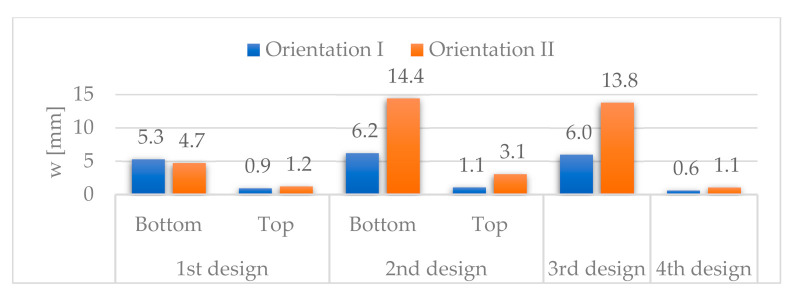
Vertical deflections, w.

**Figure 28 materials-14-06645-f028:**
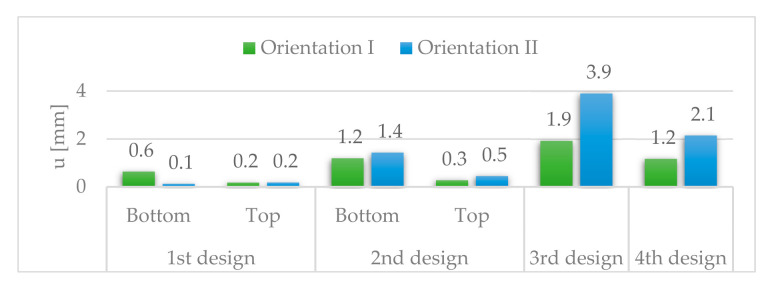
Longitudinal deflections, u.

**Figure 29 materials-14-06645-f029:**
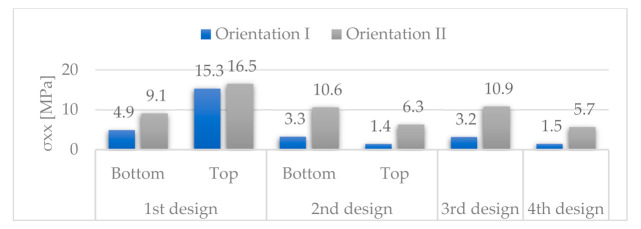
Stresses *σ**_xx_*.

**Figure 30 materials-14-06645-f030:**
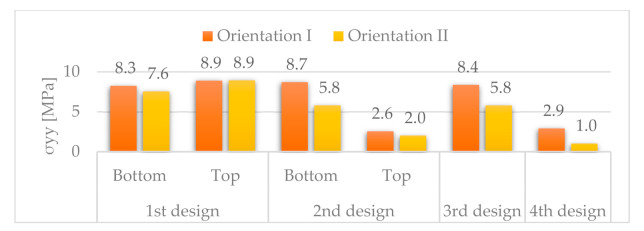
Stresses *σ**_yy_*.

**Figure 31 materials-14-06645-f031:**
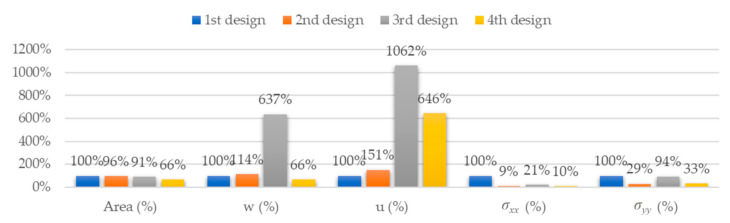
Comparative results for the best configurations found for each design.

**Table 1 materials-14-06645-t001:** Common flute types [15,23]. Reprinted with permission from ref. [12]. Copyright 2021 Elsevier.

Designation	Picture	Height (in)	Height (mm)	Flutes/m	Pitch (mm)	Take-Up Factor
A flute	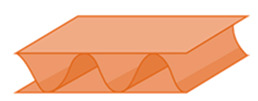	1/4″	4.8	108 ± 10	8.0–9.5	≈1.50
B flute	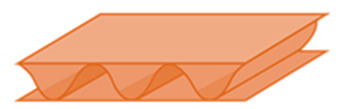	1/8″	3.2	154 ± 10	5.5–6.5	≈1.40
C flute	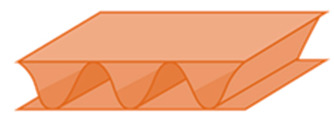	11/64″	4.0	128 ± 10	6.8–7.9	≈1.45
E flute	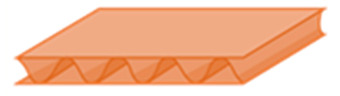	1/16″	1.6	295 ± 13	3.0–3.5	≈1.25
F flute	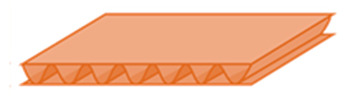	1/32″	0.8	420 ± 13	1.9–2.6	≈1.25

**Table 2 materials-14-06645-t002:** Material properties: elastic moduli, E_i_, shear moduli, G_ij_, and Poisson ratios, ν_ij_. Reprinted with permission from ref. [12]. Copyright 2021 Elsevier.

Parameter	Unit	Heavy Duty		
		Outer Liner	Inner Liner	Fluting
E_1_	MPa	8250	8180	4500
E_2_	MPa	2900	3120	4500
E_3_	MPa	2900	3120	3000
G_23_	MPa	70	70	35
G_13_	MPa	7	7	3.5
G_12_	MPa	1890	1950	1500
ν_12_	-	0.43	0.43	0.40
ν_13_	-	0.01	0.01	0.01
ν_23_	-	0.01	0.01	0.01
t	mm	0.75	0.40	0.25
h	mm	-	-	4.8
P	mm	-	-	8.5

**Table 3 materials-14-06645-t003:** Elements of the stiffness matrix for each layer, in Voigt notation.

Q_ij_	Unit	Outer Liner	Inner Liner	Fluting
Q_11_	[MPa]	8824.2	8801.4	146.2
Q_12_	[MPa]	1334.2	1444	59.807
Q_13_	[MPa]	44.361	48.01	145.44
Q_22_	[MPa]	3102	3357.2	361.6
Q_23_	[MPa]	35.71	39.08	59.755
Q_33_	[MPa]	2900.5	3120.6	146.14
Q_44_	[MPa]	70	70	4.5198
Q_55_	[MPa]	7	7	0.90365
Q_66_	[MPa]	1890	1950	5.9147

**Table 4 materials-14-06645-t004:** Comparative results for the best configurations found for each design, for α = 80°.

Design	Area (m^2^)	w (mm)	u (mm)	*σ_xx_* (MPa)	*σ_yy_* (MPa)
1st	0.87	0.9	0.2	15.3	8.9
2nd	0.83	1.1	0.3	1.4	2.6
3rd	0.79	6.0	1.9	3.2	8.4
4th	0.57	0.6	1.2	1.5	2.9

## Data Availability

All the raw/processed data required to reproduce these findings were presented in this manuscript.

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
