# Peer review of "Efficient Design of Thin Wall Seating Made of a Single Piece of Heavy-Duty Corrugated Cardboard"

_materials, 2021, doi:10.3390/ma14216645_

Round 1

Reviewer 1 Report

Journal name: Materials

Manuscript number: materials-1406616-peer-review-v1

Full Title: “Efficient design of thin wall seating made of a single piece of heavy-duty corrugated cardboard”

Authors: Berta Suarez, M Luisa M. Muneta, Gregorio Romero and Juan D. Sanz-Bobi

This work deals with the design of stools made of a triple-wall corrugated cardboard. Some finite element models are presented and discussed.

General comments: the paper is well written and formatted; figures are clear (even if not all in hi-res); the bibliography is adequate; the topic fits fairly well the purposes of the Journal.

The research proposed by the Authors is of interest from an engineering standpoint and, as far as the knowledge of the Reviewer, the results are new and here published for the first time. Nevertheless, the following questions need to be carefully addressed by the Authors prior to publication.

1) Lines 9: “tripe-wall” should read “tripLe-wall”.

2) Paragraph “Abstract”: Authors are invited to specify how their work fits into the existing technical literature and what their original contributions are.

3) Paragraph “Introduction”: thin-walled manufacts have been extensively studied in the technical literature for their buckling and warping problems. The Authors are invited to mention these issues in the “Introduction”, even discussing if such phenomena can be of interest in their design process. Suggested papers focusing on the topics are the following: https://doi.org/10.2140/jomms.2006.1.1479, for a theoretical framework, https://doi.org/10.1007/s11012-021-01349-9, for some experimental studies.

4) Line 80: multiple citations should read [22, 23], instead of [22], [23]. Please check throughout the document.

5) Section “2.1 Homogenization approach”: it is not clear if the approach is novel or not. Please discuss.

6) General remark: the Reviewer’s opinion is that, in its current form, the contribution is more a “case study paper” than an “original paper”.

Authors are invited to submit a rebuttal version of the manuscript; the Reviewer’s decision is “Major Revision”.

Reviewer 3 Report

Dear Authors, 

the manuscript "Efficient design of thin wall seating made of a single piece of heavy-duty corrugated cardboard" has a good quality. Please consider my suggestions:

The Abstract: Please follow the aim of your research, discuss more deeply the results of your research, please add one sentence about implications for practice and for future research.

The Introduction:

I suggest starting with material, then workpiece, and then methods - start with corrugated cardboard, then Thin-wall furniture, then FEA and Homogenization techniques, please consider.

Please add more discussion about the use of corrugated cardboard, as it is the main topic of your manuscript, please check here:

doi.org/10.1155/2014/654012

doi.org/10.3390/ma14175064

10.1016/S0263-8223(03)00131-4

Lines 112-159: This part belongs to the Materials and Methods section.

I suggest moving Math in lines 173-218 to Appendix.

In the Materials and Methods section I also suggest starting with Material, then Methods.

The Result part is well written, but discussion with other authors is missing, please check for example here:

doi.org/10.1016/j.compstruct.2021.113642

doi.org/10.1016/j.compstruct.2008.04.008

or many others.

Please add implications for practice and further research to the Conclusion section.

General: The topic is very current, the manuscript is well written, after fixing some issues it can be published.

Round 2

Reviewer 1 Report

The rebuttal version of the paper is eligible for publication "as is".

Reviewer 3 Report

The manuscript was significantly improved.